# A Hybrid Discrete Artificial Bee Colony Algorithm Based on Label Similarity for Solving Point-Feature Label Placement Problem

Wen Cao [1], Jiaqi Xu [1], Yong Zhang [2,*], Siqi Zhao [1], Chu Xu [1] and Xiaofeng Wu [2]

1    School of Geoscience and Technology, Zhengzhou University, Zhengzhou 450001, China; zzdx_edifier@zzu.edu.cn (W.C.); xujiaqi666@gs.zzu.edu.cn (J.X.); zzu_zsq@gs.zzu.edu.cn (S.Z.); xc1024@gs.zzu.edu.cn (C.X.)
2    Zhengzhou Zhonghe Jing Xuan Information Technology Co., Ltd., Zhengzhou 450001, China; journeyroad@163.com
*    Correspondence: zy765@163.com

**Abstract:** The artificial bee colony algorithm (ABC) is a promising metaheuristic algorithm for continuous optimization problems, but it performs poorly in solving discrete problems. To address this issue, this paper proposes a hybrid discrete artificial bee colony (HDABC) algorithm based on label similarity for the point-feature label placement (PFLP) problem. Firstly, to better adapt to PFLP, we have modified the update mechanism for employed bees and onlooker bees. Employed bees learn the label position of the better individuals, while onlooker bees perform dynamic probability searches using two neighborhood operators. Additionally, the onlooker bees' selection method selects the most promising solutions based on label similarity, which improves the algorithm's search capabilities. Finally, the Metropolis acceptance strategy is replaced by the original greedy acceptance strategy to avoid the premature convergence problem. Systematic experiments are conducted to verify the effectiveness of the neighborhood solution generation method, the selection operation based on label similarity, and the Metropolis acceptance strategy in this paper. In addition, experimental comparisons were made at different instances and label densities. The experimental results show that the algorithm proposed in this paper is better or more competitive with the compared algorithm.

**Keywords:** artificial bee colony algorithm; point-feature label placement; label similarity; Metropolis acceptance strategy

## 1. Introduction

Combinatorial optimization problems are one of the most challenging and widely used mathematical problems today. Dealing with combinatorial optimization problems means choosing the optimal solution to maximize or minimize the objective function under the given conditions. There exist many practical application scenarios for this problem, such as graph coloring problems [1], workshop scheduling [2], traveling salesman problems [3,4], and label placement problems [5]. Solving such problems is significant in terms of productivity improvement and cost reduction. Label placement is one of the most attractive branches of discrete combinatorial optimization problems. In practical applications, its solution size is usually very large, and the problem solution grows exponentially with the problem dimension, which is a typical Non-deterministic Polynomial-hard (NP-hard) problem [6]. Label placement can be understood as assigning labels to each feature on the map according to cartographic rules and preferences while ensuring maximum freedom from conflict, and ultimately obtaining a clear, beautiful, and easy-to-read map. According to the type of map, features can be divided into three different kinds of labeling problems: point features [7] (hospitals, travel spots, etc.), line features (rivers, roads, etc.) [8], and area features [9] (continents, countries, oceans, etc.) [9]. Since all three types of problems

can be abstracted as combinatorial optimization problems according to the label candidate model and the label quality evaluation function, the number of labels for point features is the largest. Thus, the most extensive research has been conducted on the placement of point-feature labels.

The difficulties of point element label placement are label conflict, label feature conflict, and label correlation. In addition, the difficulty of solving PFLP increases exponentially with the size of the problem. Current methods for solving the PFLP can be solved by both exact and metaheuristic algorithms, while exact algorithms are only suitable for small-scale optimization problems and are extremely time-consuming in solving large-scale problems. The metaheuristic algorithm can obtain optimal or near-optimal solutions in an acceptable time and is a general heuristic strategy [10]. The rules of the metaheuristic algorithm can use the current search information to adjust the search and form an intelligent iterative search mechanism. Such rules can effectively avoid falling into local optima, improve search efficiency, and have better efficiency and applicability compared to exact algorithms for solving complex optimization problems, and have become the mainstream method for solving the PFLP, such as simulated annealing [11,12], tabu search algorithms [13], genetic algorithms [14], etc. Metaheuristic algorithms fall into two main categories: single solution-based and population-based approaches. Population-based approaches are divided into two categories: evolutionary algorithms and swarm intelligence algorithms. The most common of the single solution-based approaches are tabu search and simulated annealing. Alvim and Taillard [15] used POPMUSIC to divide the problem into subproblems solved separately using tabu search. Rabello [16] combined a clustering search algorithm with simulated annealing to solve the point-feature label placement problem. Araujo et al. [17] improved Rabello's clustering search algorithm by proposing a density clustering search using three methods: density-based clustering (DBSCAN), natural group identification (NGI), and label propagation (LP) to detect promising solutions. Guerine [18] combined data mining techniques and clustering search to achieve faster convergence and better label results than the previous clustering search. Cravo et al. [19] applied the greedy adaptive random algorithm to the point-feature label placement for solving. In terms of evolutionary algorithms, genetic algorithms, and differential evolutionary algorithms are two typical examples, and Lu [20] proposed a differential evolution and genetic algorithm for the multi-geographic feature label placement problem. Deng [21] improved the differential and genetic algorithm for Lu in three aspects: selection of candidate positions, evaluation of label quality, and sequential iteration. Li et al. [22] combined genetic algorithm and tabu search to solve the point-feature label placement problem. The metaheuristic algorithm for swarm intelligence is less applied in the label placement, and only the ant colony algorithm [23] is applied to solve the point-feature label placement, mainly because most swarm intelligence algorithms are used to solve continuous optimization problems, and some improvements need to be made in solving discrete problems.

This paper focuses on an innovative application of the artificial bee colony (ABC) algorithm for point-feature label placement, which is an excellent swarm intelligence optimization algorithm for solving continuous optimization problems [24]. The renewal of the solution cooperates with other bees, and the onlooker bees expand their reinforcement capacity, while the scout bees ensure their diversity, with fewer parameters and easier implementation. The algorithm was originally proposed to solve complex sequential problems and outperformed many other algorithms when tested on complex mathematical benchmarks. The method was subsequently well applied to similar combinatorial optimization problems such as the traveling salesman problem [25,26] and job-shop scheduling [27], so we believe that the method is equally well suited to solve the point-feature label placement problem. Since this method was originally used to solve continuous problems and some improvements are needed to apply it to combinatorial optimization problems, we propose a hybrid discrete artificial bee colony algorithm (HDABC) based on label similarity to solve the point-feature label placement problem. In HDABC, a new solution update method was designed for employed bees and onlooker bees to avoid the loss of population diversity due

to a single update formula. In addition, a combination of label diversity and the roulette selection method was used to select more promising solutions for optimization in the selection phase of the onlooker bees. In addition, the greedy acceptance strategy was replaced by the Metropolis acceptance strategy to further improve the balance between exploration and diversity. The effectiveness of this paper's algorithm is verified by comparing it with other metaheuristics on the tested instances. Our main contributions are as follows:

1. A new discrete optimization algorithm (HDABC) is proposed for solving the point-feature label placement problem;
2. The update methods for employed and onlooker bees are redesigned to suit the point element label placement problem. The hired bee uses learning from good individuals, and the scout bee searches alternately with two search operators based on dynamic probabilities;
3. Onlooker bees select more promising solutions for updating based on label similarity to improve the performance of the algorithm;
4. Replace the greedy acceptance strategy of employed bees and onlooker bees with the Metropolis acceptance strategy to avoid the premature convergence problem.

The text continues here.

## 2. Point-Feature Label Placement Problem

Point-feature label placement refers to the assignment of label text to point features on the map according to certain rules such as minimization of conflicts, label preference, and non-ambiguity. The point-feature label placement based on the label candidate model and the label quality evaluation function can be abstracted into a combinatorial optimization problem. In the following, we focus on a brief description of the label candidate model and the label quality evaluation function.

### 2.1. Label Candidate Model

The point-feature label placement requires a label candidate model to provide the label position for it, and the merit of the label candidate model directly affects the result of point-feature label placement, so it is important to select a suitable label model. Label candidate models are mainly divided into fixed models [28] and sliding models [29]. The fixed model can make full use of the label gap area by the sliding strategy but the computational complexity is larger. The common fixed models are 4-orientation and 8-orientation models. Zhou et al. [23] proposed a multi-level multi-orientation model. To fully utilize the blank area of the label, multiple label orientations can be employed, but this approach may increase the time required. To balance quality and efficiency, the label candidate model used in this paper adopts eight orientations. As shown in Figure 1, the shaded part of the figure is the point element symbol, the point feature symbol cannot exceed the minimum outer circle of the set point feature, the rectangular area 0–7 represents the candidate position of the point, each position has a priority size, the positive right side is the optimal position, and the smaller the number represents its higher priority. The dashed rectangular box in the figure is the smallest external rectangle containing the point feature and the label candidate rectangle.

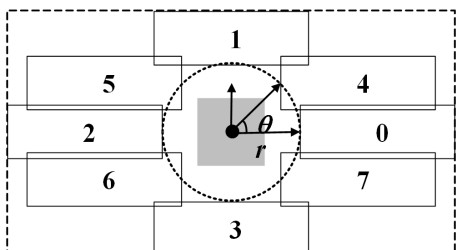

**Figure 1.** Multi-level and multi-orientation label candidate model (The numbers 0–7 represent label positions).

*2.2. Label Quality Evaluation Function*

The point-feature label placement aims to obtain conflict-free, aesthetically pleasing, and ambiguity-free point-feature label maps, and the quality evaluation function of point-feature label placement is constructed mainly considering conflict, label priority, and label correlation [30,31]. The label quality evaluation function for solution x is as follows. Label quality evaluation function abstracts the PFLP problem into a minimization objective function problem. Smaller function values indicate that the solution is better.

$$\min f(x) = \sum_{i=1}^{n} \sum_{j=1}^{m} Q(i,j)\alpha_{ij} \times 1000/n \tag{1}$$

$$Q(i,j) = \omega_1 Q_{con}(i,j) + \omega_2 Q_{pri}(i,j) + \omega_3 Q_{ass}(i,j) \tag{2}$$

Constraints:

$$\sum_{j=1}^{m} \alpha_{ij} = 1 \tag{3}$$

$n$ is the number of point features, $m$ is the number of label candidates, $Q(i,j)$ is the evaluation function when the $i$-th point feature is at the $j$-th label candidate position, $\alpha_{ij}$ is the switch variable, when the $i$-th point-feature label is at the $j$-th label candidate position, $\alpha_{ij} = 1$, and vice versa $\alpha_{ij} = 0$. $Q_{con}(i,j)$ is the conflict score between the $i$-th point feature and the other point features when the $i$-th point feature is located at the $j$-th candidate position. It is set to 1 if there is a conflict with other points; otherwise, it is set to 0. $Q_{pri}(i,j)$ is the label priority score when the $i$-th point feature is located at the $j$-th label candidate position. Using the example of the 8 orientations label candidate model, $Q_{pri}(i,j)$ is assigned as follows: the rightmost direction is 0, the top direction is 1/8, the leftmost direction is 2/8, the bottom direction is 3/8, the top-right direction is 4/8, the top-left direction is 5/8, the bottom-left direction is 6/8, and the bottom-right direction is 7/8. $Q_{ass}(i,j)$ is the label relevance score when the $i$-th point feature is located at the $j$-th label candidate position. $\omega_1, \omega_2, \omega_3$ represent the weight proportion of conflict, label priority, and label correlation, respectively, which usually take the values of 0.5, 0.3, 0.2. Since label correlation is related to the minimum distance that can be recognized by the human eye, the ambiguity distance cannot be accurately measured. Therefore, in this paper, we do not consider the label correlation and set it to 0.

## 3. A Hybrid Discrete Artificial Bee Colony Algorithm

*3.1. Standard Artificial Bee Colony Algorithm*

The artificial bee colony (ABC) algorithm is a swarm intelligence optimization algorithm derived from the social behavior of honey bees and is used to solve numerical optimization problems. The solution to the problem to be optimized represents the location of the food source, and the amount of nectar in the food source represents the adaptation value of the corresponding solution. The artificial bee colony algorithm consists of a combination of employed bees, onlooker bees, and scout bees, with equal numbers of employed and observation bees. The algorithm is divided into four phases, namely: initialization phase, employed bee phase, onlooker bee phase, and scout bee phase.

1.  Initialization phase

Randomly initialize the population $p$, consisting of a total of $N$ individuals, each of which is a $d$-dimensional vector. It is constructed as follows:

$$x_{ij} = l_j + r(u_j - l_j) \tag{4}$$

where $x_{ij}$ is the $j$th dimension of the $i$th solution, $i \in \{1, 2, \ldots N\}$ and $j \in \{1, 2, \ldots d\}$. $N$ is the population size. $l_j$ represents the lower and $u_j$ represents the upper bound of the parameter $x_{ij}$, and $r$ is a random number between [0, 1].

2.  Employed bee phase

Employed bees play a vital role in searching for food sources, gathering information on their location and quality, and sharing this information with other bees. Each employed bee is assigned to a specific food source, meaning that there are an equal number of employed bees and food sources. Neighborhood search can be performed according to Equation (5) to generate new solutions to find better food sources.

$$v_{ij} = \begin{cases} x_{ij} & if\ j \neq q \\ x_{ij} + \varphi(x_{ij} - x_{kj}) & else \end{cases} \tag{5}$$

$x_i$ represents the food source to be updated, $x_k$ is a randomly selected food source, and $v_i$ is a newly generated food source. $v_{ij}$ corresponds to the $j$th dimension of $v_i$, $x_{ij}$ is the $j$th dimension of $x_i$, and $x_{kj}$ is the $k$th dimension of $x_k$. $i \in \{1, 2, \dots N\}$, $k \in \{1, 2, \dots N\}$, $j \in \{1, 2, \dots d\}$, and $k \neq i$. $q$ is a randomly chosen dimension and $q \in \{1, 2, \dots d\}$. $\varphi$ is a random number between $[-1, 1]$. After producing the new candidate food source $v_i$, its label quality evaluation function is calculated. Then, a greedy selection is applied between $v_i$ and $x_i$. At this stage, each solution has the opportunity to be improved.

3.  Onlooker bee phase

When all the employed bees have completed their search, they share information about the food source in the dance area, and the onlooker bees evaluate the information provided by the employed bees to select the food source by roulette. The higher the adaptation value of the nectar source, the greater the probability of selection by the onlooker bees. The adaptation value is calculated as follows, and the selection probability is given in Equation (7), with N being the number of employed bees. After selecting a food source (xi), the onlooker bee performs a search to generate a new solution based on Equation (5) and greedily accepts the new solution. Some solutions may receive multiple opportunities for improvement during this phase, while some solutions may not have the opportunity for improvement.

$$fit(x_i) = \frac{1}{f(x_i)} \tag{6}$$

$$p_i = \frac{fit(x_i)}{\sum\limits_{i=1}^{N} fit(x_i)} \tag{7}$$

where $f(x_i)$ is the label quality evaluation function value of the $i$th solution, and $fit(x_i)$ is the fitness of $x_i$. Since the label quality evaluation function is a minimization problem, smaller values indicate better solutions. Therefore, it is necessary to take the inverse of $f(x_i)$ to get $fit(x_i)$ for roulette selection. The larger the $fit(x_i)$ value, the better the solution. $p_i$ is the choice probability of the $i$th solution.

4.  Scout bee phase

During this phase, scout bees are used to find new food sources not found by the employed bees and onlooker bees, avoiding the search process from falling into a local optimum. When the quality of the solution exceeds the set number of searches limit L, the solution is considered to be fully explored and the employed bee is transformed into a scout bee that uses Equation (4) to generate a random solution to replace the current solution. This is why onlooker bees choose the better solution for exploration with higher probability.

*3.2. A Hybrid Discrete Artificial Bee Colony Algorithm Based on Label Similarity*

3.2.1. Coding and Initialization

In the algorithm proposed in this paper, a real number encoding is used to represent the solution of PFLP. In this paper, we have chosen an 8-orientation label candidate model, where the 8 label candidate positions of each point feature are encoded using numbers from 0 to 7. In addition, the standard artificial bee colony algorithm uses Equation (4) to initialize

the colony; however, PFLP is a typical discrete optimization problem, and Equation (4) is no longer suitable. Therefore, in this paper, the swarm is initialized randomly. The initial swarm is generated by randomly selecting candidate positions from the eight label candidate positions of the point features.

### 3.2.2. Generation of Neighborhood Solutions

The standard ABC solution update equation is used to solve continuous optimization problems. For discrete combinatorial optimization problems like PFLP, the update equation needs to be redesigned to accommodate PFLP. First is the redefinition of subtraction, assuming that $e_i$ and $g_i$ are the $i$-th dimension of the solutions $e$ and $g$, respectively.

$$e_i - g_i = \begin{cases} g_i \text{ ,if } e_i \neq g_i \text{ and } f(e) > f(g) \\ rand, else \end{cases} \tag{8}$$

The standard ABC algorithm is essentially learning from other solutions and therefore defines subtraction as learning the label positions of other solutions. If the learned label position is different from the original and the learned bee has a better fitness, the label position of the better bee is learned, and vice versa, a random label position is given. Combined with the above subtraction operation, the updated formula of the solution is shown in Equation (9), and the addition indicates replacing the original label position value in the $i$-th dimension of $e$ with the newly learned label position value.

$$e = e + (e_i - g_i) \tag{9}$$

However, if both employed bees and onlooker bees update the label position by Equation (9), the way of updating the label position is too single to fully explore the solution space. Therefore, we integrate some transformation operators into the proposed algorithm for neighborhood solution generation, and use Equation (9) for the employed bees to update, while two operators, shift and conflict-shift, are used for the onlooker bees to enhance the search. The generation of neighboring solutions can result in changes in the label positions of points, all of which may potentially lead to a reduction in conflicts and changes in label priorities, thereby decreasing the label quality evaluation function.

Shift: A point $p_i$ is randomly selected from 1-$n$ dimensions and a new label position is randomly generated instead of the original one. This operator is the most commonly used operator to generate new solutions for point-feature label placement. The shift operator updates only one dimension, and the update of one dimension guarantees that the solution space is explored at a finer granularity. Figure 2 is a schematic diagram of the shift neighborhood transformation operator. There are a total of 8 points, and the point at index 2 is selected to transform the label position 5 into the label position 4.

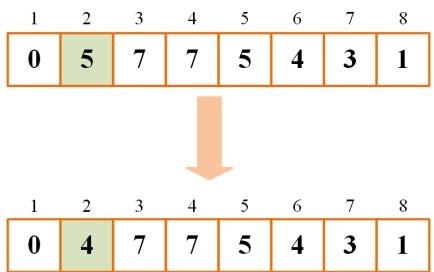

**Figure 2.** Schematic diagram of the shift neighborhood transformation operator (The value on the box is the index of the point, the value in the box is the label position).

Conflict-shift: randomly select a point $p_i$ from 1-$n$ dimensions, determine the set $C$ of point features that conflict with the point, and generate a new label position of $p_i$ and all points in the set $C$ to replace the original position. The conflict-shift operator is more perturbed for the update of the solution compared to the shift operator, which can

effectively jump out of the local optimum. Figure 3 is a schematic diagram of the conflict-shift transformation operator. There are eight points in total. Select the point with index 4. Compute its set of conflicting point features $C$. The point features in $C$ are indexed 1 and 3. Randomly generate label orientations to replace the original positions for point features indexed 1, 3, 4 in an attempt to eliminate conflicts between point features.

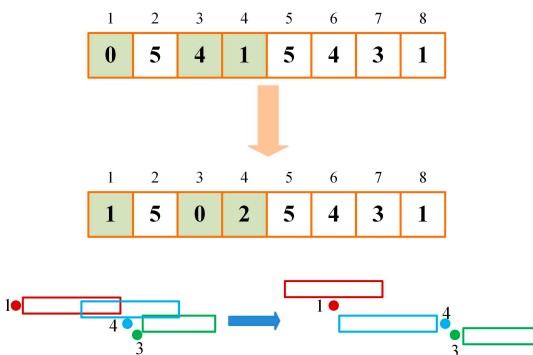

**Figure 3.** Schematic diagram of the conflict-shift neighborhood transformation operator (The value on the box is the index of the point, the value in the box is the label position).

For neighborhood solution generation, a higher degree of perturbation is required to generate neighborhood solutions at the beginning of the iteration to help improve the quality of the solution quickly and jump out of the local optimum, while a higher perturbation later in the iteration may fail to produce a more optimal solution. This is because the solution is very close to the approximate global optimum late in the iteration, and a larger perturbation will move the solution away from the approximate global optimum. Therefore, for the two operators, we use the same probability of random selection for use in the initial stage, and the conflict-shift is more perturbing for the solution, and we keep reducing the selection probability of the conflict-shift operator and increasing the selection probability of the shift operator in the iterative process, but the probability of the conflict-shift operator cannot be lower than a certain threshold. In summary, the selection probability of the operator should be adjusted dynamically with the number of iterations of the algorithm, which is an adaptive parameter, and the dynamic probabilities of the two operators are as follows:

$$D_{cs} = \min_{cs} + \left(\frac{\max_{cs} - \min_{cs}}{iter_{\max}}\right)(iter_{\max} - iter) \tag{10}$$

$$D_c = 1 - D_{cs} \tag{11}$$

where $D_{cs}$ is the dynamic selection probability of the conflict-shift operator, $D_c$ is the selection probability of the shift operator, $iter_{\max}$ is the maximum number of iterations, $iter$ is the current number of iterations, and $\max_{cs}$ and $\min_{cs}$ are the maximum and minimum selection probabilities of the conflict-shift operator.

### 3.2.3. Selection Operation Based on Label Similarity

Onlooker bees are used to enhance their search capabilities in standard ABC algorithms, where roulette selection is commonly used to select food sources associated with employed bees. However, the difference between the fitness values of each solution is not very large, so the difference in the selection probability between each solution is not large. The key to ABC is that the neighborhood of good solutions has a higher chance of finding a better solution compared to the neighborhood of poor solutions, and more exploration of good solutions is needed. Therefore, the roulette selection method cannot cause selection pressure and thus weaken the performance of the artificial bee colony algorithm. To improve the performance of the artificial bee colony algorithm, we select the

most promising solutions to generate neighborhood solutions based on the label similarity between individual solutions to enhance the search capability of the onlooker bees. We classify solutions into superior and inferior food solutions according to the mean size of their fitness values. Those with fitness values below the mean are classified as superior solutions, and vice versa as inferior solutions. Based on the solution information shared by the employed bees, if the food source is inferior, it will visually inspect the surrounding food sources based on a certain detection probability $p_b$ to select the most suitable food source, i.e., the onlooker bees will evaluate similar food sources to select the best one. The method puts more improvements on more promising solutions effectively improving the algorithm search capability. Since the encoding of our solutions is the label position of each point, we can choose the Hamming distance to measure the similarity between solutions. Use Equations (12) and (13) to measure the similarity between the solutions $x_i$ and $x_j$.

$$dv_{ij}(k) = \begin{cases} 1, if\ S_i[k] = S_j[k] \\ 0, else \end{cases} \tag{12}$$

$$d(i,j) = \sum_{k=1}^{n} dv_{ij}(k) \tag{13}$$

where $d(i,j)$ is the similarity of solutions $x_i$ and $x_j$, $n$ is the dimension of the point-feature label placement, $k$ is the $k$th dimension of the solution, $d$ is the similarity of solutions $x_i$ and $x_j$ in the $k$th dimension, and $S_i[k]$, $S_j[k]$ are the label positions of solutions $x_i$ and $x_j$ in the $k$th dimension, respectively. Label positions are abstracted into a 0–7 encoding.

Figure 4 gives a simple example of the label similarity measure. In this example, a PFLP problem with dimension 10 is given. The similarity generated by each dimension of $x_i$ and $x_j$ is calculated according to Equation (12). If the label position of the same dimension is the same then set it to 1, otherwise set it to 0. Finally, the total similarity degree $d(i,j) = 2$ is calculated according to Equation (13).

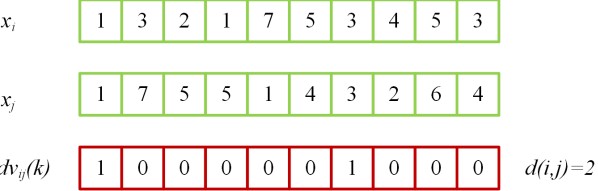

**Figure 4.** Schematic diagram of label similarity calculation (The numbers 0–7 represent label positions).

The selection operation based on the label similarity (SOLS) is specified as follows: firstly, the food sources are classified into superior food sources and inferior food sources based on the fitness of the solutions. Assume that $x_i$ is the solution obtained by the roulette selection method selection, and if the selected solution is an inferior food source, the similarity between $x_i$ and the remaining solutions is calculated with a certain detection probability $p_b$ to solve its similarity. Then the solutions are ranked according to their similarity from largest to smallest, and the top $p$ number of solutions are selected and added to the candidate table of that solution. If there is a solution in the candidate table with fitness less than $x_i$, the best solution $x_l$ in that solution set is used instead of the solution $x_i$ obtained from roulette, which is updated with dynamic probability using two operators.

### 3.2.4. Metropolis Acceptance Strategy

In the standard ABC algorithm, both employed bees and onlooker bees use a greedy strategy to decide whether to accept a new solution. However, in solving combinatorial optimization problems like PFLP, greedy acceptance strategies usually lead to premature convergence problems due to their discrete character. Therefore, this paper uses the Metropolis acceptance criterion to determine whether to accept poor solutions to increase the diversity of solutions. Suppose $f(A)$ is the evaluation function of the current solution $A$

and $f(B)$ is the evaluation function of the new solution $B$. If the new solution is better than the old solution, the new solution directly replaces the old solution, and vice versa, the Metropolis acceptance criterion is used to determine whether to accept the poor solution. The probability formula for accepting the poor solution is as follows:

$$p = \begin{cases} 1, if\ f(B) < f(A) \\ e^{-(f(B)-f(A))/T}, else \end{cases}, T = T \times \alpha \tag{14}$$

where $T$ is the current temperature and $\alpha$ is the cooling parameter, which usually requires setting an initial temperature $T_0$ that is continuously cooled down during the iterative process. As the number of iterations reaches the predefined annealing length ($SA_{max}$), the temperature undergoes a reduction. Cooling continues until the specified minimum temperature ($T_{min}$) is attained, at which point the cooling process ceases.

### 3.2.5. Reset the Scout Bee

In a standard ABC, when there is no improved bee solution within a certain number of times, the employed bee abandons the nectar source to become a scout, and the scout randomly generates a new solution to replace the original one. However, for PFLP, randomly generating a new solution to the scout bees is not a good strategy because the way the new solution of PFLP is generated is not suitable for improving the randomly generated new solution quickly. Thus, we try to use t times multiple conflict-shift operators on the solution to be dropped. Try to jump out of the current stagnant state using the conflict-shift operator.

The overall flowchart of the hybrid discrete ABC algorithm based on label similarity is shown in Figure 5.

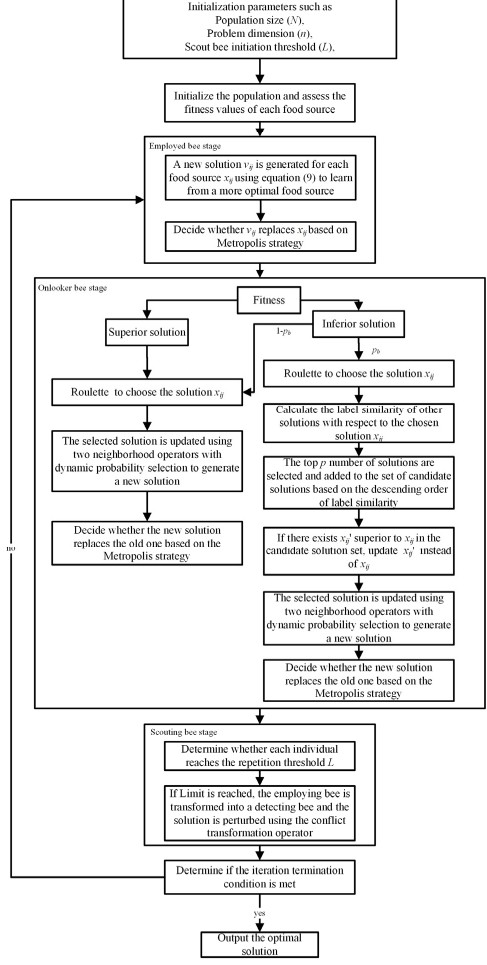

**Figure 5.** The overall flowchart of the hybrid discrete ABC algorithm based on label similarity.

## 4. Experimental Results and Analysis

This section focuses on presenting the experimental results of the HDABC based on label similarity. We compare the performance of our algorithm with some existing algorithms in the literature to evaluate the effectiveness of HDABC. Furthermore, we analyze and discuss several essential components of HDABC.

### 4.1. Instance and Parameter Settings

To verify the performance of HDABC, we employed web crawling techniques to retrieve POI data from both Kaifeng and Beijing in China. A portion of this data was then selected to create actual point feature datasets consisting of 103, 266, and 525 points, respectively. We compared HDABC with genetic algorithm (GA) [14], simulated annealing (SA) [32], tabu search (TS) [13], and discrete differential evolution and genetic algorithm (DDEGA) [21]. The specific experimental parameters are listed in Table 1. The maximum number of iterations for HDABC, GA, TS, and DDEGA is 10,000. The iteration ends when the algorithm reaches the maximum number of iterations or the optimal solution reaches 2000 repetitions. For SA, the iteration ends when the maximum temperature drops to the minimum temperature, or when the number of repetitions of the optimal solution reaches 30,000. When the specified maximum number of repetitions is reached, the algorithm continues to iterate without any changes. The symbol size $r$ and the distance from the coordinate point to the label are 5 and 10 pixels, respectively, and the label font is 12 pixels. All methods are implemented by Microsoft Visual Studio in C++, and experiments were carried out on an Intel (R) Core (TM) i5-8500 3.0 GHz processor with 8 GB of RAM.

**Table 1.** Parameter setting.

| Algorithm | Parameter | Parameter Definition | Value |
|---|---|---|---|
| HDABC | $N$ | Number of employed bees and onlooker bees | 50 |
| | $L$ | Scout bee activation threshold | 500 |
| | $T_0$ | Initial temperature | 1 |
| | $\alpha$ | Cooling speed | 0.95 |
| | $T_{min}$ | minimum temperature | 0.01 |
| | $NP$ | Population size | 100 |
| | $p$ | Candidate table size | 4 |
| | $p_b$ | Detection probability | 0.5 |
| GA | $SA_{max}$ | Annealing length | 20 |
| | $p_m$ | Elite Probability | 0.1 |
| | $p_e$ | Crossover probability | 0.8 |
| | $p_v$ | Mutation probability | 0.1 |
| | $T_0$ | Initial temperature | 40,000 |
| | $\alpha$ | Cooling speed | 0.95 |
| | $SA_{max}$ | Annealing length | 4000 |
| SA | $T_c$ | Termination temperature | 0.01 |
| | $m_v$ | Label transformation probability | 0.001 |
| | $cn$ | Conflict count | / |
| | $CL$ | Candidate Table Size | $2 + 0.2 \times n$ |
| TS | $TL$ | Contraindicated table size | $5 + 0.2 \times cn$ |
| | $p_{DE}$ | Weight of DE | 0.7 |
| | $p_{GA}$ | Weight of GA | 0.3 |
| DDEEGA | $F$ | DE variation probability | 0.5 |
| | $C_r$ | DE hybridization probability | 0.8 |
| | $C_{GA}$ | Genetic variation probability | 0.1 |
| | $NP$ | Population size | 100 |

### 4.2. Experimental Results and Comparison with Other Algorithms

To verify the effectiveness of the proposed algorithm in this paper, HDABC was compared with GA, SA, TS, and DDEGA. The experiment was conducted with eight commonly used label densities: 5%, 10%, 15%, 20%, 25%, 30%, 35%, and 40%. Label density $\rho$ refers to the ratio of the sum of the symbol and label area in the map to the total map area, reflecting the density of feature and label distribution. The comparison is mainly conducted from two aspects: the number of labels and the label quality evaluation function. To ensure the reliability of the results, each algorithm was independently run 10 times and the data was averaged. Tables 2 and 3 present the comparison of the number of labels without conflicts between HDABC and GA, SA, TS, and DDEGA for label densities from 5% to 40%. A higher number of conflict-free labels indicates a better result. Tables 4 and 5 show the comparison of the evaluation function of label quality between HDABC and GA, SA, TS, and DDEGA for label densities from 5% to 40%. In this comparison, a smaller value indicates a better result. In these tables, "Instance" represents the test case, where $I_1$, $I_2$, and $I_3$ correspond to acquired test cases with 103, 266, and 525 test cases, respectively. "Algorithm" denotes the algorithm used, and "Best" indicates the best result obtained from ten runs for each algorithm. "Average" represents the average result obtained from the ten runs for each algorithm. Figure 6 shows the ranking graph of the average label number and quality evaluation function for each algorithm.

In terms of the number of labels, Tables 2 and 3 show that the differences between algorithms are not significant for the 103-point dataset, and in some cases, the average and best numbers of labels are the same. The label situation for small datasets is relatively simple, so the differences between algorithms are not significant. However, as the dataset size increases, the differences between algorithms become apparent. For the 266-point and 506-point datasets, the proposed algorithm achieved more average and best numbers of labels compared to GA, TS, and DDEGA. Compared to SA, HDABC outperformed SA in terms of the average and best numbers of labels in most cases, and only slightly underperformed SA in certain instances and label densities. Overall, HDABC provided higher quality solutions for PFLP for the vast majority of test cases and label densities in terms of the number of labels. Additionally, Figure 6a displays the rankings of various algorithms based on label number, with lower values indicating superior performance of the algorithm. As shown in Figure 6a, HDABC had the highest overall rank, followed by SA, TS, and GA, while DDEGA had the lowest rank.

**Table 2.** Comparison of the number of labels for HDABC and other algorithms under 5–20% label densities.

| Instance | Algorithm | $\rho = 5\%$ | | $\rho = 10\%$ | | $\rho = 15\%$ | | $\rho = 20\%$ | |
| --- | --- | --- | --- | --- | --- | --- | --- | --- | --- |
| | | Best | Average | Best | Average | Best | Average | Best | Average |
| | HDABC | 88 | 87 | 84 | 83 | 79 | 78 | 77 | 77 |
| | GA | 88 | 87 | 84 | 82 | 78 | 77 | 77 | 76 |
| $I_1$ | SA | 88 | 87 | 84 | 83 | 79 | 78 | 77 | 76 |
| | TS | 88 | 87 | 84 | 83 | 79 | 78 | 77 | 76 |
| | DDEGA | 88 | 86 | 83 | 82 | 79 | 77 | 77 | 76 |
| | HDABC | 213 | 211 | 189 | 186 | 172 | 170 | 157 | 154 |
| | GA | 210 | 208 | 185 | 180 | 168 | 163 | 153 | 147 |
| $I_2$ | SA | 212 | 210 | 190 | 185 | 172 | 169 | 157 | 153 |
| | TS | 212 | 210 | 187 | 185 | 169 | 166 | 153 | 150 |
| | DDEGA | 206 | 205 | 182 | 177 | 163 | 160 | 149 | 145 |
| | HDABC | 438 | 435 | 398 | 395 | 366 | 363 | 347 | 341 |
| | GA | 433 | 429 | 382 | 387 | 356 | 352 | 333 | 327 |
| $I_3$ | SA | 437 | 434 | 397 | 393 | 366 | 363 | 344 | 341 |
| | TS | 436 | 434 | 397 | 392 | 366 | 361 | 341 | 338 |
| | DDEGA | 425 | 422 | 377 | 381 | 351 | 344 | 323 | 317 |

**Table 3.** Comparison of the number of labels for HDABC and other algorithms under 25–40% label densities.

| Instance | Algorithm | $\rho = 25\%$ | | $\rho = 30\%$ | | $\rho = 35\%$ | | $\rho = 40\%$ | |
|---|---|---|---|---|---|---|---|---|---|
| | | Best | Average | Best | Average | Best | Average | Best | Average |
| $I_1$ | HDABC | 77 | 75 | 75 | 74 | 71 | 69 | 68 | 66 |
| | GA | 76 | 74 | 74 | 71 | 69 | 67 | 66 | 63 |
| | SA | 77 | 75 | 74 | 73 | 70 | 68 | 67 | 65 |
| | TS | 76 | 74 | 74 | 72 | 70 | 68 | 67 | 65 |
| | DDEGA | 73 | 71 | 69 | 72 | 68 | 65 | 64 | 62 |
| $I_2$ | HDABC | 147 | 144 | 138 | 134 | 128 | 125 | 123 | 119 |
| | GA | 141 | 137 | 134 | 127 | 121 | 118 | 115 | 110 |
| | SA | 147 | 144 | 139 | 134 | 129 | 125 | 120 | 117 |
| | TS | 144 | 141 | 134 | 130 | 123 | 120 | 118 | 114 |
| | DDEGA | 138 | 134 | 127 | 124 | 120 | 116 | 115 | 109 |
| $I_3$ | HDABC | 322 | 317 | 301 | 294 | 283 | 276 | 263 | 259 |
| | GA | 306 | 300 | 282 | 278 | 267 | 260 | 248 | 242 |
| | SA | 320 | 316 | 297 | 293 | 279 | 275 | 267 | 261 |
| | TS | 316 | 313 | 296 | 290 | 277 | 272 | 261 | 256 |
| | DDEGA | 297 | 291 | 279 | 272 | 261 | 253 | 245 | 238 |

**Table 4.** Comparison of the label quality evaluation functions for HDABC and other algorithms under 5–20% label densities.

| Instance | Algorithm | $\rho = 25\%$ | | $\rho = 30\%$ | | $\rho = 35\%$ | | $\rho = 40\%$ | |
|---|---|---|---|---|---|---|---|---|---|
| | | Best | Average | Best | Average | Best | Average | Best | Average |
| $I_1$ | HDABC | 153.6 | 157.6 | 163.2 | 169.1 | 187.1 | 190.4 | 201.4 | 206.7 |
| | GA | 154.4 | 161.4 | 171.1 | 177.1 | 195.4 | 200.1 | 204.7 | 213.9 |
| | SA | 152.1 | 158.7 | 167.0 | 172.6 | 188.6 | 194.7 | 199.5 | 205.8 |
| | TS | 161.0 | 167.3 | 173.2 | 180.2 | 192.7 | 196.3 | 205.7 | 209.1 |
| | DDEGA | 161.7 | 169.6 | 175.7 | 181.6 | 193.6 | 203.4 | 214.2 | 217.7 |
| $I_2$ | HDABC | 246.6 | 249.3 | 266.2 | 268.1 | 280.8 | 285.1 | 294.8 | 297.7 |
| | GA | 256.0 | 260.3 | 268.7 | 278.7 | 290.5 | 296.5 | 304.8 | 310.4 |
| | SA | 245.3 | 250.3 | 264.4 | 268.1 | 278.4 | 285.1 | 296.1 | 297.6 |
| | TS | 250.3 | 255.6 | 272.7 | 276.8 | 289.9 | 293.9 | 300.9 | 305.0 |
| | DDEGA | 258.1 | 263.0 | 276.8 | 281.7 | 293.0 | 298.3 | 307.0 | 311.8 |
| $I_3$ | HDABC | 217.5 | 221.5 | 240.3 | 242.1 | 256.5 | 259.7 | 269.9 | 273 |
| | GA | 231.0 | 235.6 | 253.6 | 257.2 | 269.1 | 273.3 | 283.2 | 289.3 |
| | SA | 219.4 | 221.9 | 241.8 | 244.3 | 257.1 | 260.3 | 269.1 | 272.6 |
| | TS | 222.5 | 225.1 | 244.5 | 247.1 | 260.2 | 263.7 | 274.6 | 278.2 |
| | DDEGA | 232.3 | 237.5 | 255.2 | 259.4 | 271.1 | 275.1 | 286.8 | 288.9 |

Based on Tables 4 and 5, it can be seen that in the majority of cases, HDABC outperforms other algorithms in terms of both the average label quality evaluation function and the best label quality evaluation function, regardless of the label and instances. At the 5% label density scenario of the 103-point dataset, HDABC, GA, SA, and DDEGA all achieved good results in terms of the optimal label quality evaluation function. As the dataset size and label density increase, the label placement becomes more complex and dense, and the advantages of the proposed algorithm become more apparent. The algorithm provides a higher quality solution for PFLP for the vast majority of instances and label density scenarios. In terms of the average label quality, HDABC is only inferior to SA at the 40% label density, while in all other scenarios, it is superior or equal to all other algorithms. From the perspective of the best label quality, HDABC is slightly inferior to SA in only a few cases, while in all other scenarios, it is superior or equal to all other algorithms. Figure 6b displays the rankings of various algorithms based on the label quality evaluation function, with lower values indicating superior performance of the algorithm. According to

Figure 6b, HDABC has the highest ranking in terms of performance across various datasets and label densities, followed by SA, TS, and GA, with DDEGA having the lowest ranking.

**Table 5.** Comparison of the label quality evaluation functions for HDABC and other algorithms under 25–40% label densities.

| Instance | Algorithm | $\rho = 25\%$ | | $\rho = 30\%$ | | $\rho = 35\%$ | | $\rho = 40\%$ | |
| --- | --- | --- | --- | --- | --- | --- | --- | --- | --- |
| | | Best | Average | Best | Average | Best | Average | Best | Average |
| $I_1$ | HDABC | 153.6 | 157.6 | 163.2 | 169.1 | 187.1 | 190.4 | 201.4 | 206.7 |
| | GA | 154.4 | 161.4 | 171.1 | 177.1 | 195.4 | 200.1 | 204.7 | 213.9 |
| | SA | 152.1 | 158.7 | 167.0 | 172.6 | 188.6 | 194.7 | 199.5 | 205.8 |
| | TS | 161.0 | 167.3 | 173.2 | 180.2 | 192.7 | 196.3 | 205.7 | 209.1 |
| | DDEGA | 161.7 | 169.6 | 175.7 | 181.6 | 193.6 | 203.4 | 214.2 | 217.7 |
| $I_2$ | HDABC | 246.6 | 249.3 | 266.2 | 268.1 | 280.8 | 285.1 | 294.8 | 297.7 |
| | GA | 256.0 | 260.3 | 268.7 | 278.7 | 290.5 | 296.5 | 304.8 | 310.4 |
| | SA | 245.3 | 250.3 | 264.4 | 268.1 | 278.4 | 285.1 | 296.1 | 297.6 |
| | TS | 250.3 | 255.6 | 272.7 | 276.8 | 289.9 | 293.9 | 300.9 | 305.0 |
| | DDEGA | 258.1 | 263.0 | 276.8 | 281.7 | 293.0 | 298.3 | 307.0 | 311.8 |
| $I_3$ | HDABC | 217.5 | 221.5 | 240.3 | 242.1 | 256.5 | 259.7 | 269.9 | 273 |
| | GA | 231.0 | 235.6 | 253.6 | 257.2 | 269.1 | 273.3 | 283.2 | 289.3 |
| | SA | 219.4 | 221.9 | 241.8 | 244.3 | 257.1 | 260.3 | 269.1 | 272.6 |
| | TS | 222.5 | 225.1 | 244.5 | 247.1 | 260.2 | 263.7 | 274.6 | 278.2 |
| | DDEGA | 232.3 | 237.5 | 255.2 | 259.4 | 271.1 | 275.1 | 286.8 | 288.9 |

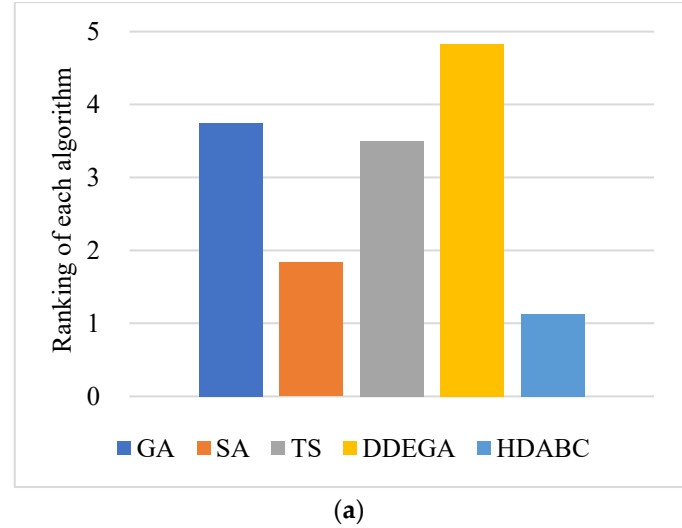

(**a**)

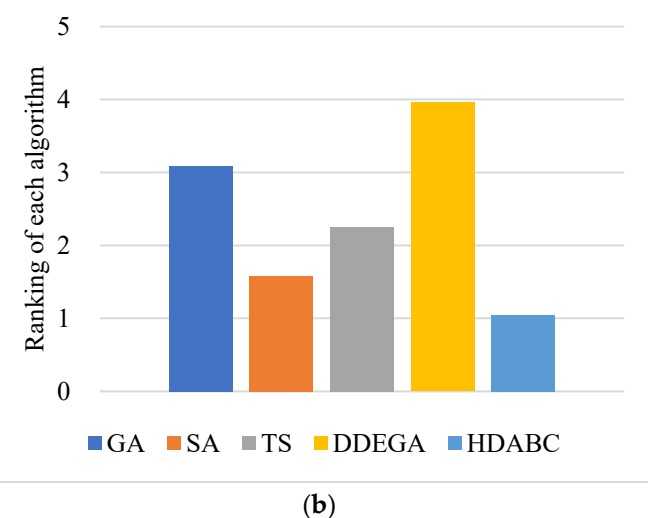

(**b**)

**Figure 6.** Ranking of each algorithm: (**a**) Ranking of each algorithm based on label number; (**b**) Ranking of each algorithm based on label quality evaluation function.

*4.3. Analysis and Discussion*

4.3.1. Analysis of Neighborhood Solution Generation

In the traditional artificial bee colony algorithm, employed bees and onlooker bees mainly generate new solutions by Equation (5), which is too single and not well suited for discrete problems. Therefore, we use Equation (9) for the employed bee for updating and two search operators for the onlooker bee with dynamic probability to better fit the discrete problem of PFLP. To verify the effectiveness of the proposed neighborhood solution generation, we compare the proposed neighborhood solution generation method with Equation (5) in this paper, and the results are shown in Figure 7. The results indicate that the HDABC update approach for different instances and densities produces significantly better results than the traditional ABC solution update method using Equation (5), with an average reduction of 20.6 in the average label quality. The employed bees learning

from good food sources is consistent with the features of PFLP while maintaining the good properties of the original update equation. The dynamic and alternating use of two neighborhood operators by the onlooker bees provides a certain amount of randomness to the algorithm and effectively avoids getting stuck in local optima traps.

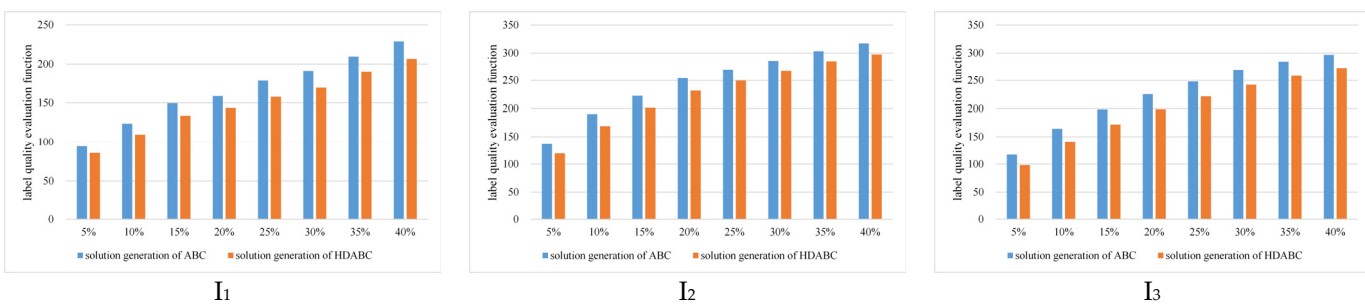

**Figure 7.** Comparison of solution generation between ABC and HDABC.

4.3.2. The Role of Selection Operations Based on Label Similarity

The main purpose of the ABC algorithm is to enhance the development ability of the onlooker bees, so a suitable selection operation is needed. The traditional selection operation is roulette selection, and we compare the selection operation based on the label similarity (SOLS) proposed in this paper with roulette selection. The specific comparison results are shown in Table 6. Firstly, in terms of label quality scores, the selection operation based on label similarity performs better than the original roulette wheel selection in most instances and label densities. Moreover, on the 525-point dataset, the average label quality of the selection operation based on label similarity is superior to that of the roulette wheel selection. The selection based on label similarity updates the solution with more promising ones, leading to better label results in most instances and label densities. Additionally, we can observe that in the case of 103 data, the running time of the selection operation based on label similarity is similar to that of the roulette wheel selection method. However, as the size of the dataset increases, the running time of the selection operation based on label similarity decreases by 45.1% compared to the roulette wheel selection method. As the size of the dataset increases, the difficulty of solving the point-feature label problem also increases. The termination condition of the iteration in this study is when the optimal solution remains unchanged for a certain number of iterations or reaches the maximum number of iterations. Compared to the roulette wheel selection method, the selection operation based on label similarity provides more opportunities for more promising solutions, thus accelerating the convergence speed of the algorithm.

4.3.3. The Role of Metropolis Acceptance Strategy

In the standard artificial bee colony algorithm, the employed bees and onlooker bees accept new solutions through a greedy acceptance criterion. However, this acceptance strategy can quickly lead to premature convergence in discrete problems. Therefore, we replaced the greedy acceptance criterion with the Metropolis acceptance criterion to avoid this issue. We then compared the traditional artificial bee colony algorithm with the greedy acceptance criterion and the honey bee dance algorithm with the Metropolis acceptance criterion. The specific results are shown in Figure 8. The results indicate that the Metropolis acceptance strategy is significantly better than the greedy acceptance strategy, with an average decrease of 14.6 in average label quality. This suggests that the acceptance of lower quality solutions facilitated by the Metropolis acceptance criterion enhanced the algorithm's ability to escape local optima and effectively avoided issues related to premature convergence. Furthermore, the Metropolis acceptance criterion was able to achieve a good balance between exploration and exploitation.

**Table 6.** Comparison of selection operation based on label similarity and roulette wheel selection.

| Instance | $\rho$ | SOLS | | | Roulette Selection | | |
|---|---|---|---|---|---|---|---|
| | | Best | Average | Time | Best | Average | Time |
| $I_1$ | 5% | 84.5 | 85.9 | 25 | 84.5 | 86.1 | 27 |
| | 10% | 108.6 | 109.8 | 29 | 108.6 | 111.5 | 27 |
| | 15% | 132.9 | 133.9 | 26 | 132.9 | 135.8 | 25 |
| | 20% | 141.5 | 143.6 | 28 | 141.5 | 143 | 29 |
| | 25% | 153.6 | 157.6 | 28 | 153.2 | 156.7 | 30 |
| | 30% | 163.2 | 169.1 | 31 | 166.6 | 170.6 | 33 |
| | 35% | 187.1 | 190.4 | 31 | 186.2 | 189.1 | 33 |
| | 40% | 201.4 | 206.7 | 27 | 203.3 | 205.5 | 30 |
| $I_2$ | 5% | 116.8 | 119.6 | 73 | 116.5 | 118.7 | 158 |
| | 10% | 166.5 | 169.1 | 71 | 167.7 | 170.5 | 133 |
| | 15% | 199.2 | 201.2 | 60 | 200.8 | 202.2 | 130 |
| | 20% | 228.1 | 231.7 | 75 | 230.7 | 232.8 | 124 |
| | 25% | 246.6 | 249.3 | 68 | 247 | 249.7 | 107 |
| | 30% | 266.2 | 268.1 | 56 | 264.9 | 267.7 | 115 |
| | 35% | 280.8 | 285.1 | 62 | 282.1 | 284.6 | 115 |
| | 40% | 294.8 | 297.7 | 58 | 294.9 | 297.9 | 111 |
| $I_3$ | 5% | 96.0 | 98.2 | 183 | 97 | 98.4 | 413 |
| | 10% | 139.3 | 141.1 | 200 | 139.2 | 142.3 | 370 |
| | 15% | 170 | 172 | 173 | 170.5 | 172.2 | 358 |
| | 20% | 196.6 | 198.4 | 222 | 195.5 | 198.6 | 351 |
| | 25% | 217.5 | 221.5 | 166 | 220.9 | 223 | 313 |
| | 30% | 240.3 | 242.1 | 204 | 243.4 | 244.8 | 299 |
| | 35% | 256.5 | 259.7 | 188 | 259.1 | 261.3 | 300 |
| | 40% | 269.9 | 273 | 172 | 271.2 | 274.5 | 287 |

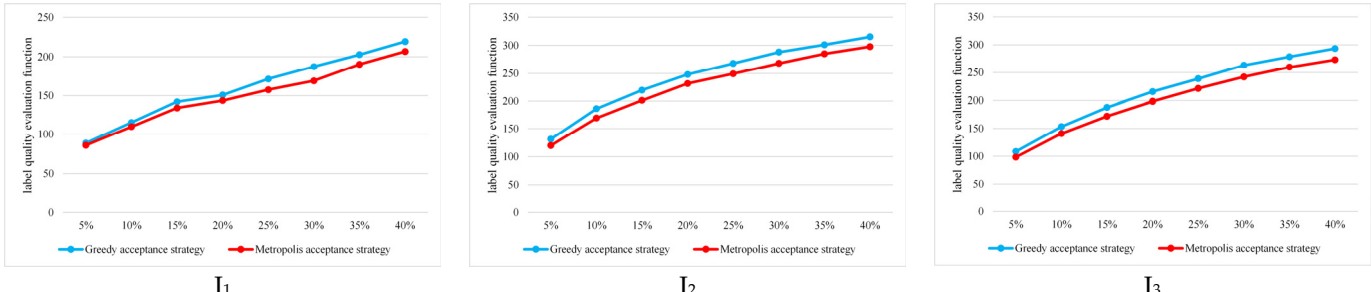

**Figure 8.** Comparison between the greedy acceptance strategy and the Metropolis acceptance strategy.

## 5. Conclusions and Future Outlook

In this paper, we propose a hybrid discrete artificial bee colony algorithm based on label similarity to solve point-feature label placement problems. Originally designed for continuous problems, we adapted some steps of the ABC algorithm to better suit discrete problems. Specifically, the neighborhood solution generation was modified by introducing a learning mechanism in the employed bees and dynamic probability-based use of two neighborhood search operators for onlooker bees. The selection operation was improved by label similarity to identify more promising solutions for updating. Lastly, the Metropolis acceptance criterion was implemented in place of the original greedy acceptance criterion for a better balance between exploration and exploitation. To validate the effectiveness of our method, we compared it with other algorithms on various instances and label densities. The experimental results demonstrate that our approach is a qualified and competitive solution for point-feature label placement problems. In future work, we will devise a more reasonable ambiguity factor in the label quality evaluation function and explore the application of ABC to other combinatorial optimization problems.

**Author Contributions:** Wen Cao and Jiaqi Xu designed the research; Wen Cao and Jiaqi Xu conceived the experiments; Wen Cao and Jiaqi Xu conducted the experiments and analyzed the results; Wen Cao and Jiaqi Xu contributed to the drafting of the work; Wen Cao, Jiaqi Xu, Yong Zhang, Siqi Zhao, Chu Xu and Xiaofeng Wu contributed to the review and editing of the manuscript. All authors have read and agreed to the published version of the manuscript.

**Funding:** This research received no external funding.

**Data Availability Statement:** Not applicable.

**Conflicts of Interest:** The authors declare that the research was conducted in the absence of any commercial or financial relationships that could be construed as a potential conflict of interest.

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
