# Peer review of "A Hybrid Discrete Artificial Bee Colony Algorithm Based on Label Similarity for Solving Point-Feature Label Placement Problem"

_ijgi, doi:10.3390/ijgi12100429_

Round 1
Reviewer 1 Report
Dear colleagues,
thank you for your work on a hybrid discrete artificial bee colony algorithm based on label similarity for solving point-feature label placement.
As cartographer this problem is a very important one and as the introduction describes, this problem´s complexity increases with the amount and variety of features (point, lines, areas).
From my point of view, the approach selected is an interesting one. The compilation of existing algorithms for searching in the introduction gives a good insight into the landscape of algorithms in use.
In line 35, you make use of an appreviation, which is not explained.
In figure 7, you use Chinese letters for the image description.
In general, the paper was hard to follow in the beginning. Maybe a visual expample of an use casde could help to get the reader into the topic.
As cartographer I look forward to pragmatic examples of this set of algorithms for the creation of maps and their labelling.
Best regards
Author Response
We feel great thanks for your professional review work on our article. As you are concerned, several problems need to be addressed. According to your nice suggestions, we have made extensive corrections to our previous manuscript, the detailed corrections are listed in the annex.

Reviewer 2 Report
In the paper “A hybrid discrete artificial bee colony algorithm based on label similarity for solving point-feature label placement problem” by Wen Cao et. al. the authors present the artificial bee colony algorithm for point feature label placement
The method, described in the paper is very interesting, but unfortunately, it is not at all clear from the text how this method works.
1) The authors often mention “better solution”, but nowhere is the criterion on what basis one or another solution is considered “better”.
2) What determines the boundaries x_min and x_max and how do these values change during the transformation of bees?
3) The right side of formula (4) does not depend on i, while the left side does. One gets the impression that all x_ij take the same value for all i=1,…,N at a fixed j. A similar remark to formula (5). The right side depends on k, but the left side does not.
4) In formula (8) it is not clear why in the first line e_i-f_i=f_i, although fitness(e_i)> fitness(f_i), In this regard, the question of the criterion of a “better solution” arises again.
5) It is necessary to explain the physical meaning of the designations fit_i and f_i in formulas (6) and (7).
6) How is the number of iterations chosen?
7) Figure 2 repeats part of Figure 2. Both of these figures represent a completely incomprehensible scheme. There is no explanation in the text.
8) How are the values recorded in tables 2-5 obtained? How can you understand from these numbers which method is better?
9) What is plotted on the vertical axis of the histograms in Figures 6 and 7? How do these histograms imply that ABC is worse than HDABC?
10) Line 447-449 says: "Compared to the roulette wheel selection method, the selection operation based on label similarity provides more opportunities for more promising solutions, thus accelerating the convergence speed of the algo". How does this follow from the presented results?
11) In Figure 7, some of the captions are in Chinese.
12) Line 208" Since an 8 orientation” is probably a typo.
13) Some simplest example is necessary to demonstrate where the algorithm can be applied.
I think that the article cannot be accepted for publication until the authors write a text that is understandable to the reader.
Author Response

(The authors gave the same response as above.)

Reviewer 3 Report
I believe Table 1 could have more lines so any reader could align which parameters and values are aligned to which algorithm.
Author Response

(The authors gave the same response as above.)

Round 2
Reviewer 2 Report
After the revision of the paper A hybrid discrete artificial bee colony algorithm based on label 2 similarity for solving point-feature label placement problem by Wen Cao et.al., many points were clarified, the motivation of the work and its difference from other works on artificial bee colony algorithms were marked. However, the mathematical part is still not entirely clear.
1) What does v_{ij} mean in formula (5)? Is this a new resource? Is it contained in the set x_{ij} or is it newly generated?
2) how is the function f(x) determined in formula (1), from which the minimum is then taken?
3) How should formula (9) be understood? Which component of the solution “e” are we talking about?
4) Figures 2 and 3 are again unclear. Before Figure 2 it is written: “The point at index 2 is selected to transform the label orientation 5 into the label orientation 4”. How was this replacement planned in the bee colony? Why did the remaining cells change? Why in the first cell the value 0 is first replaced with the value 1, and then back to 0? Can the sources change their quality evaluation function?
5) What does the term “label orientation” mean in formula (12)?
6) Why there is the value 1 in the last column of the red line in Figure 4 when the values x_i and x_j are not the same?
I think that all explanations should be added to the mathematical part before the article is accepted for publication in IJGI.
Author Response
We feel great thanks for your professional review work on our article. As you are concerned, several problems need to be addressed. According to your nice suggestions, we have made extensive corrections to our previous manuscript, the detailed corrections are in the annex.

Round 3
Reviewer 2 Report
Now everything is clear. I think that the article can be published in IJGI